# Ten-Second Cold Water Stress Test Differentiates Parkinson’s Disease from Multiple System Atrophy: A Cross-Sectional Pilot Study

**DOI:** 10.3390/biomedicines13071585

**Published:** 2025-06-28

**Authors:** Makoto Takahashi, Wataru Hagiwara, Sakiko Itaya, Keisuke Abe, Tetsuya Maeda, Akira Inaba, Satoshi Orimo

**Affiliations:** 1Department of Neurology, Kanto Central Hospital, Tokyo 158-8531, Japan; aimuatoydarian@gmail.com (W.H.); sakiko8757@gmail.com (S.I.); abekei1015@gmail.com (K.A.); a-inaba@kanto-ctr-hsp.com (A.I.); sorimo0307@gmail.com (S.O.); 2Division of Neurology and Gerontology, Department of Internal Medicine, School of Medicine, Iwate Medical University, Morioka 028-3694, Japan; maeda@iwate-med.ac.jp; 3Kamiyoga Setagaya Street Clinic, Tokyo 158-0098, Japan

**Keywords:** Parkinson’s disease, multiple system atrophy, cold stress test, thermography, surface temperature

## Abstract

**Background/Objectives**: Patients with Parkinson’s disease (PD) often have cold hands and experience frostbite. The diagnostic criteria for multiple system atrophy (MSA) also describe cold and discolored hands; however, in our clinical experience, the hands are relatively warm. These symptoms are thought to be caused by autonomic dysfunction; however, the detailed mechanisms and differences in cold hands between MSA and PD remain unclear. We aimed to identify an appropriate cold stimulation test to differentiate patients with PD and MSA using finger surface temperature (FST). **Methods**: We included a total of 34 patients, 27 with PD and 7 with MSA diagnosed at least 5 years after disease onset. After 15 min in a room with constant temperature and humidity, the patient’s hand was placed in cold water at 4 °C for 10 s as the cold water stress test (10sec-CWST). FST was captured using a thermal imaging camera every minute for 15 min, and the recovery of FST was analyzed. The association between the clinical characteristics of each patient and the degree of FST recovery was examined. **Results**: All patients completed the 10sec-CWST without adverse events. Patients with PD showed a significantly slower recovery of FST after 7 min compared to those with MSA, with a maximum difference at 11 min (PD: 8.1 ± 0.6 °C; MSA: 10.5 ± 0.3 °C; *p* < 0.01). FST recovery at 11 min was negatively correlated with the degree of resting hand tremor (r = −0.585, *p* < 0.01). **Conclusions**: FST after 10sec-CWST may be a safe and efficient test to differentiate PD and MSA.

## 1. Introduction

Parkinson’s disease (PD) is the second most common neurodegenerative disorder, with an incidence of 10–18 per 100,000 person-years [1]. PD is characterized by the loss of dopaminergic neurons in the substantia nigra, causing extrapyramidal symptoms, including bradykinesia, resting tremor, and rigidity [1,2]. Furthermore, the autonomic nervous system (ANS) is disturbed in PD, and constipation, a common autonomic symptom, is often observed in the early onset or before motor symptoms develop [1]. Multiple system atrophy (MSA) is a neurodegenerative disorder with an incidence rate of 0.49 per 100,000 person-years, which is lower than PD [3]. MSA is characterized by central autonomic dysfunction, extrapyramidal symptoms with poor levodopa responsiveness, or cerebellar ataxia [4]. The parkinsonian variant of MSA (MSA-P), in which parkinsonian symptoms predominate, is clinically similar to PD [5]. Differentiation between PD and MSA is based on the presence of olfactory disturbance [6] and cerebellar ataxia [4], timing of the onset of falls, speed of progression, and abnormal findings on head magnetic resonance imaging, including the hot-cross bun sign and atrophy of the putamen; however, it is often difficult to differentiate between the two diseases, especially in the early stages of onset.

Patients with PD and MSA show autonomic symptoms even in the early phase; however, the degree and pattern of autonomic failure differ clinically and pathologically between these two diseases, and these differences provide clues for differentiation. Pathologically, the autonomic disturbance in PD is predominantly present in the peripheral ANS, whereas that in MSA is predominantly present in the central ANS [7]. Cardiac metaiodobenzylguanidine (MIBG) scintigraphy, which indicates the degree of peripheral cardiac sympathetic denervation [8], shows a strong decrease in accumulation in PD, whereas almost no decrease in accumulation in MSA is shown until the advanced stage [9,10]. Clinically, patients with MSA show more severe orthostatic hypotension compared to those with PD [11]. In terms of urinary disturbance, PD is primarily associated with urinary frequency, whereas MSA is often associated with urinary retention [4,12].

A large number of patients with PD exhibit cold hands and feet, which suggests autonomic dysfunction [13], and some patients also experience chilblains during winter. Furthermore, the diagnostic criteria for patients with MSA include “cold, discolored hands and feet” as a supportive non-motor feature [4]. However, the percentage of patients presenting with cold hands is low, equating to 6.3% in PD and 20.3% in MSA [5]. We have also observed that patients with MSA tend to have relatively warm hands, even during winter, in our daily practice experience. Cold hands and feet are thought to be caused by autonomic disturbances; however, the precise mechanisms and differences in cold hands between PD and MSA are unclear. In recent years, the diagnosis and evaluation of various diseases, such as Raynaud’s phenomenon, rheumatoid arthritis, and carpal tunnel syndrome, have been performed by measuring hand temperature using thermography [14]. Furthermore, several studies have focused on differentiating PD and MSA by hand temperature measured using thermography, with or without cold stimulation [15,16]; however, no consensus has been reached to date. Based on the above, we hypothesized that patients with MSA would exhibit faster hand temperature recovery to cold stimulation compared to those with PD; therefore, we decided to perform the cold stimulation test.

Prior to this study, we conducted a pilot study in which healthy participants cooled their hands with an ice pack for 2 min. We found that it caused severe pain and that the cooling area was uneven. Therefore, the purpose of the current study was to develop a simple, safe, and reproducible cold stimulation test to differentiate patients with PD and MSA and to investigate whether there are differences in finger surface temperature (FST) between the two diseases following the cold stress test.

## 2. Materials and Methods

### 2.1. Participants

We included consecutive patients with a clinical diagnosis of PD or MSA who were admitted to Kanto Central Hospital and provided consent for the test. The recruitment period was from 15 September 2016 to 31 December 2018. The clinical diagnoses of PD and MSA were made according to the Movement Disorder Society clinical diagnostic criteria for PD [17] and the second consensus statement on the diagnosis of MSA [18], respectively. Patients who underwent the 10 s cold water stress test, as noted below, but whose diagnosis changed within 5 years of onset, were excluded from the analysis. We also excluded patients with comorbidities, such as diabetes mellitus, endocrine disorders, intracranial diseases, chronic heart failure, myocardial infarction, and peripheral neuropathies, or medications, such as hormones, autonomic drugs, and anti-psychotic drugs, that could affect FST. Clinical information, including age, sex, disease duration, intensity of motor symptoms using the Unified Parkinson’s Disease Rating Scale (UPDRS) part III, presence and intensity of various non-motor symptoms, accumulation of cardiac MIBG scintigraphy, levodopa dosage at enrollment, and levodopa equivalent daily dose (LEDD), was collected from medical records during hospitalization from 19 September 2016 to 12 December 2018. The most recent diagnosis was obtained from the medical record on 28 June 2024.

### 2.2. Ten-Second Cold Water Stress Test (10sec-CWST)

Examinations were performed with continued oral medications, including Levodopa. A handheld FLIR-E5 thermal imaging camera (FLIR E5; FLIR Systems, Wilsonville, OR, USA) was used to capture FST measurements. The parameters of the FLIR-E5 were as follows: sensor focal array size, 120 × 90 pixels; noise equivalent differential temperature (NETD), <0.1 °C; emissivity correction variable, 0.1 to 1.0; sampling rate, 9 Hz; and the object temperature range, −20 °C to +250 °C.

To measure FST, the patient’s hand was first photographed at an angle of approximately −10° from a distance of 1.0 m without a tripod using the FLIR-E5, and then a circular, 6 mm diameter, grossly defined region of interest immediately proximal to the middle fingernail was measured using dedicated software (FLIR tools, RRID:SCR_016330, version 5.13.17214; FLIR Systems) (Appendix A).

Before the examination, the participants spent at least 15 min relaxing in the examination room, where the temperature was kept at 25 °C and the humidity at 50%. After acclimation to the measurement environment, the temperatures of both hands were first taken as a reference temperature before cold stimulation. After taking the reference temperature, the patient’s hand on the side with the more severe akinesia was immersed in cold water at 4 °C for 10 s. After wiping off the cold water, both hands were photographed every minute for 15 min using a thermal imaging camera. The hand recovery temperature was calculated as the difference between the reference temperature before cold water stimulation and the temperature at the time of measurement.

### 2.3. Statistical Analysis

Statistical analyses were performed using EZR [19] (version 1.67; Saitama Medical Center, Jichi Medical University, Saitama, Japan). The Welch *t*-test and Mann–Whitney U-test were used to compare the PD and MSA groups. The chi-square test was used to compare categorical variables. The relationships between continuous variables were analyzed using Pearson’s correlation coefficient, and nonparametric variables were analyzed using Spearman’s rank correlation coefficient. The sensitivity and specificity of the 10sec-CWST in differentiating PD from MSA were examined by creating a receiver operating characteristic curve. Two-sided *p*-values less than 0.05 were considered statistically significant.

## 3. Results

### 3.1. Patients’ Background Data

Twenty-seven patients with PD and eight with MSA underwent the 10sec-CWST without serious adverse events. One patient with MSA was excluded from the analysis because of a change in the clinical diagnosis from MSA to progressive supranuclear palsy within 5 years of disease onset.

Patient background data are summarized in Table 1. Of the seven patients with MSA, five had cerebellar variants of MSA, and two had MSA-P. Only one patient with PD was aware of winter frostbite. The severity of the resting tremor, odor test score (odor stick identification test for the Japanese [OSIT-J]; a 12-point test to identify 12 different odors), degree of accumulation on cardiac MIBG scintigraphy, levodopa dosage, and LEDD were significantly different between patients with PD and those with MSA. FST on the test side immediately before the 10sec-CWST was 1.9 °C lower in the PD group than in the MSA group (*p* = 0.0475). No one had experienced Raynaud’s phenomenon prior to the onset of the disease.

### 3.2. Change in FST After 10sec-CWST

The 10sec-CWST cooled hands to an average temperature of 10.4 °C in patients with PD and MSA. A few patients complained of mild chills during the test. There was a statistical difference between FST at 1–3 min after the 10sec-CWST on the examined side and that on the unexamined side at 1–2 and 9–12 min (Appendix A). When comparing FST recovery temperatures, patients with PD showed a significantly slower recovery of FST than that of patients with MSA from 7 to 15 min after the 10sec-CWST, with a maximum difference at 11 min (PD: 8.1 ± 0.6 °C; MSA: 10.5 ± 0.3 °C, *p* < 0.01) (mean ± standard error) (Figure 1 and Figure 2). When the cutoff line of the recovery FST at 11 min was set at 9.6 °C, the two groups could be differentiated with a sensitivity of 70.3% and specificity of 85.7%, with an area under the curve of 0.786 (Figure 3).

### 3.3. Factors Related to the Change in FST

In the analysis of all patients, a significant correlation was found between the degree of FST recovery at 11 min, the degree of resting tremor (r = −0.585, *p* < 0.01), and the OSIT-J score (r = 0.464, *p* < 0.01) (Figure 4). When the analysis was restricted to the PD group, only the degree of resting tremor showed a statistically significant negative correlation (r = −0.474, *p* = 0.012).

## 4. Discussion

Our study had three important findings: (1) the 10sec-CWST resulted in a sufficiently cool hand uniformly and safely for patients with PD and MSA; (2) although differences were also observed in the pre-stimulus FST between the two groups, a more obvious differentiation between PD and MSA could be made by the degree of change in FST after the 10sec-CWST; and (3) the degree of resting tremor correlated with the degree of change in FST.

The first important finding was that the 10sec-CWST was performed safely and cooled the hands evenly and sufficiently, with no adverse events or dropouts in patients with PD and MSA. The commonly performed cold stress test, which involves using ice packs or cold water for more than 1 min, is painful and carries the risk of hyperventilation and dropout [16,20]. Furthermore, in cold stress tests using ice packs, the contact area between the coolant and skin is non-uniform, resulting in varying degrees of cooling in different areas. Therefore, we conducted a cold stress test with cold water to achieve uniform cooling and investigated the minimum time required for a sufficient cooling effect. In the CWST conducted on healthy participants as a pilot test, a few were able to tolerate the test for more than 30 s. A cooling of approximately 10 °C was observed after 10 s of CWST, which was considered sufficient for the study. The current study on patients with PD and MSA also showed an average cooling of 10.4 °C with the 10sec-CWST, and all patients were able to complete the examination without serious adverse effects, although some complained of a slight chill during CWST.

The second and most important finding was that patients with PD showed a significantly slower recovery of FST than those with MSA in the 10sec-CWST and that the 10sec-CWST could distinguish between these two groups based on the degree of FST recovery. The PD group also showed lower FSTs than that of the MSA group before cold stimulation; however, the difference was more pronounced in terms of recovery temperatures. A previous report showed that the palm temperature of patients with PD is lower than that of normal controls in a standard environment [15], and another report demonstrated a slower recovery of FST in patients with PD compared to healthy participants after cold stimulation [13]. The pathomechanism of these phenomena is thought to be the hypercontraction of peripheral vessels as a result of denervation hypersensitivity of the noradrenergic receptors in peripheral vessels [21]. However, some reports indicate that the vasoconstriction of peripheral vessels following cold stimulation is impaired in patients with MSA [18], with a further study showing that the vasoconstrictive ability of the skin sympathetic nerves is also weakened in patients with MSA [22]. These phenomena are thought to be caused by disturbances in the autonomic nerves of the hypothalamus or medulla. These findings suggest that the difference in FST recovery after the 10sec-CWST between the PD and MSA groups in the present study may be due to the difference in peripheral vasoconstrictor responses derived from differences in ANS disturbance. However, the current study did not examine the pathogenesis of the disease, and further research is expected.

The third finding was that the degree of FST was negatively correlated with the degree of resting tremor. Clinically, PD has been classified into three motor subtypes: tremor-dominant (TD), postural instability of gait disturbance (PIGD), and intermittent [23]. This difference is assumed to be related to the spread of Lewy bodies and the disturbance of various nervous systems other than the nigrostriatal system [24]. These subtypes are not limited to differences in motor symptoms but have also been found to be associated with a variety of non-motor symptoms. Although there have been no previous reports to our knowledge on the relationship between hand temperature and motor subtypes in PD, Gu et al. [25] studied the relationship between autonomic symptoms using the Scale for Outcomes in Parkinson’s disease for Autonomic symptoms and the motor subtypes of PD. They reported that patients with PIGD- and indeterminate-type PD have more sweating and thermoregulatory symptoms, whereas patients with TD- and indeterminate-type PD have more cardiovascular symptoms. The regulation of body surface temperature in the hands involves the contraction and dilation of peripheral blood vessels, as well as the production of perspiration. Perspiration is involved in the regulation of hand surface temperature and the vasoconstrictor response noted above. It is essential to note that the participants in this study may have had their tremors modified by dopamine replacement therapy (DRT), and it cannot be ruled out that patients with a better response to DRT may have experienced a faster recovery in FST. The present study did not investigate the mechanism underlying differences in FST recovery. The involvement of vasoconstriction and sweating and the effect of reactivity to DRT are topics for future research.

This study had some limitations. The patients in our study were clinically diagnosed, and no pathological diagnosis was made. However, the accuracy of the clinical diagnosis was increased by analyzing only patients whose diagnosis remained unchanged during more than 5 years of follow-up after disease onset. The number of patients with PD and MSA in this study was small, and there were no patients in either group who exhibited the typical discolored, cold hands or feet associated with these conditions. No healthy participants were included in this study. Because this study included only Japanese patients, we do not know whether similar findings can be obtained in other ethnic groups. However, it may be possible to extract characteristics of Japanese patients with PD and MSA.

In future studies, we aim to increase the sample size to approximately 30 cases in each group, based on the results of this study, to confirm the reliability of the findings. We would also like to observe changes over time and make comparisons with age-matched healthy subjects. In addition, to investigate the cause of the difference in hand temperature recovery between PD and MSA, we would like to examine the relationship with other autonomic nervous system symptoms and autonomic tests to better understand the pathophysiology of each disease.

## 5. Conclusions

In conclusion, the 10sec-CWST could be performed safely in patients with PD and MSA, and there were significant differences in the change in FST after the 10sec-CWST between these two groups. The 10sec-CWST may be an efficient test for differentiating patients with PD and MSA.

## Figures and Tables

**Figure 1 biomedicines-13-01585-f001:**
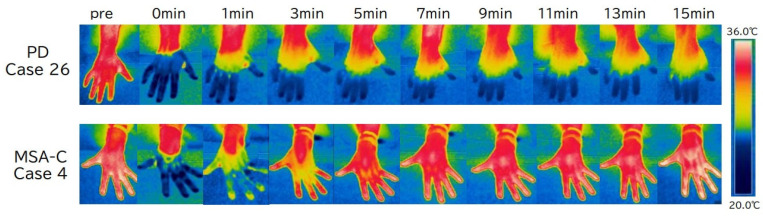
Typical images of finger surface temperature (FST) changes over time in a patient with Parkinson’s disease (PD) and one with multiple system atrophy (MSA). FST in PD (case 26) rarely recovers, while it shows rapid improvement in MSA (case 4). CWST, cold water stress test; MSA-C, multiple system atrophy—cerebellar subtype; PD, Parkinson’s disease.

**Figure 2 biomedicines-13-01585-f002:**
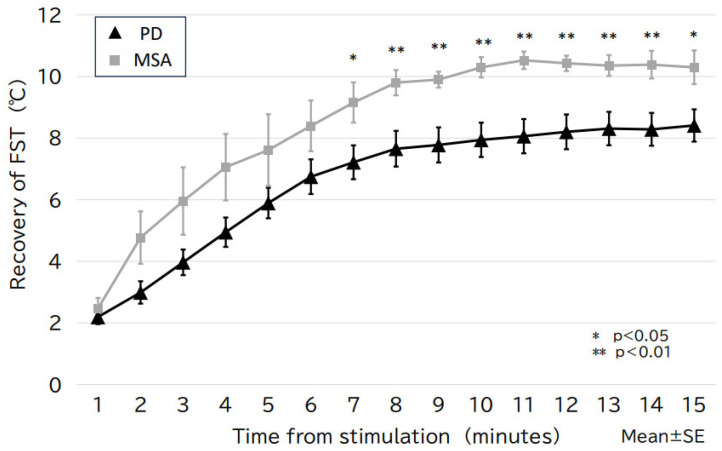
The change in FST over time for the PD and MSA groups. The PD group shows slower FST recovery than that of the MSA group, with a statistically significant difference after 7 min of stimulation (maximal difference after 11 min). FST, finger surface temperature; MSA, multiple system atrophy; PD, Parkinson’s disease; SE, standard error.

**Figure 3 biomedicines-13-01585-f003:**
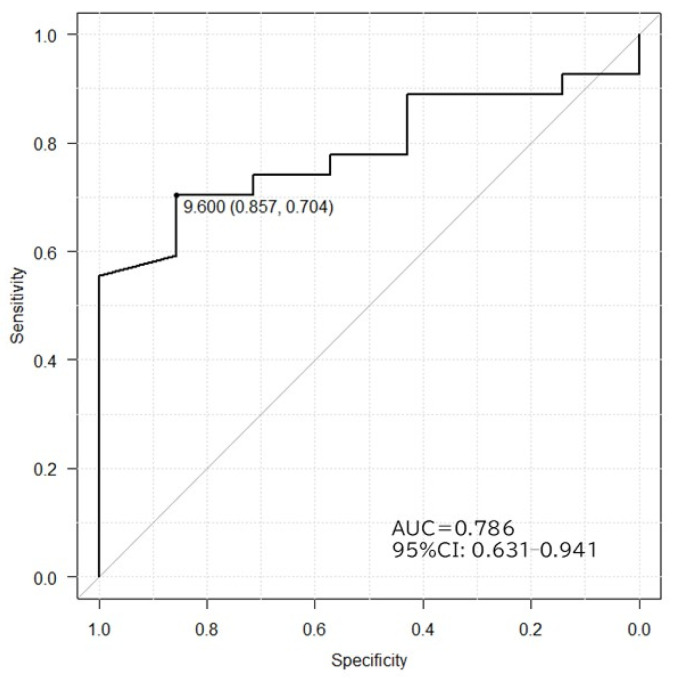
Receiver operating characteristic curves in PD and MSA differentiated by FST recovery after 11 min of stimulation. A cutoff value of 9.6 °C gives a sensitivity of 70.6%, specificity of 85.7%, and maximum area under the curve of 0.786. AUC, area under the curve; CI, confidence interval.

**Figure 4 biomedicines-13-01585-f004:**
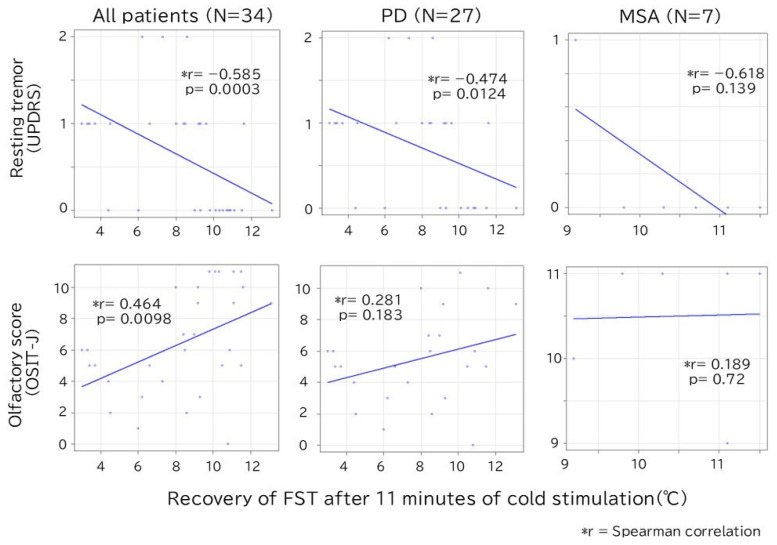
Correlation between recovery of FST after 11 min of cold stimulation and resting tremor/olfactory score. In all patients, there is a significant correlation between FST and resting tremor/olfaction scores, but only resting tremor is statistically significant in the PD group. FST, finger surface temperature; MSA, multiple system atrophy; OSIT-J, odor stick identification test for the Japanese; PD, Parkinson’s disease; UPDRS, Unified Parkinson’s Disease Rating Scale.

**Table 1 biomedicines-13-01585-t001:** Patient’s background.

	Unit	PD (*n* = 27)	MSA (*n* = 7)	*p*
			MSA-C/P (5/2)	
Male–female	N	10:17	02:05	1.000
Age (mean) (range)	year	69.4 (41–88)	61.9 (52–78)	0.088
Disease duration (mean) (range)	year	6.1 (1–17)	3 (1–5)	0.090
Hohen–Yahr stage (mean) (range)	-	2.8 (1–4)	3 (2–4)	0.604
Resting tremor * (mean) (range)	-	0.7 (0–2)	0.1 (0–1)	0.039
Rigidity * (mean) (range)	-	1.6 (1–2)	1.0 (0–2)	0.057
RBD	N	9	2	1.000
OH	N	6	4	0.157
Constipation	N	20	5	1.000
Tachyuria	N	15	2	0.398
Dysuria	N	0	4	0.157
Dyshidrosis	N	8	1	0.644
Chilblains	N	1	0	1.000
OSIT-J (mean) (range)	-	5.5 (1–11)	10.5 (9–11)	0.001
MIBG early (mean) (range)	H/M	1.92 (1.23–3.28)	3.22 (3.07–3.52)	0.009
MIBG delay (mean) (range)	H/M	1.63 (1.02–2.85)	2.76 (2.63–2.93)	0.009
L-dopa (mean) (range)	mg/day	321.3 (0–800)	78.57 (0–450)	0.029
LEDD (mean) (range)	mg/day	472.35 (0–1440)	90 (0–450)	0.018
FST (mean) (range)	°C	32.9 (26.4–36.7)	34.8(34.4–35.4)	0.048

* UPDRS score of the hand on the examining side. The severity of the resting tremor, olfactory score, degree of metaiodobenzylguanidine uptake, levodopa dosage, and levodopa equivalent daily dose are significantly different between Parkinson’s disease and multiple system atrophy. PD, Parkinson’s disease; MSA, multiple system atrophy; RBD, REM sleep behavior disorder; OH, orthostatic hypotension; OSIT-J, odor stick identification test for the Japanese; MIBG, metaiodobenzylguanidine; L-dopa, levodopa; LEDD, levodopa equivalent daily dose; FST, finger surface temperature on the examination side.

## Data Availability

The original image of the thermal imaging camera can be disclosed upon request. The FST data are presented in Appendix A.

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
