# Peer review of "Ten-Second Cold Water Stress Test Differentiates Parkinson’s Disease from Multiple System Atrophy: A Cross-Sectional Pilot Study"

_biomedicines, 2025, doi:10.3390/biomedicines13071585_

Round 1
Reviewer 1 Report
Comments and Suggestions for Authors
Ten-Second Cold Water Stress Test Differentiates Parkinson’s Disease from Multiple System Atrophy: A Pilot Study
I have read the manuscript with interest, and you can find my appraisal, section by section, as follows:
As a general remark, I hope that my commentaries and appraisal can help improve the manuscript and give more insights.
Abstract: line 17: Please clarify the total sample size and the group size. This is not completely clear.
Introduction: The section is well-written, but the symptoms of PD and MSA need to be described better at the start of the section. I also advise adding a brief description of the epidemiological data for PD and MSA, and adding related citations.
Moreover, I suggest adding the principal alterations that are possible to observe with MRI in MSA.
An important concern is about warm hands in MSA, since you have observed in your clinical practice, but you need to quantify this or report published data. Please, add.
Line 66: Since you study this phenomenon with infrared imaging, using a thermal camera, you need to improve the state of the art about previous studies that used this or a similar method/paradigm. You can find some information here: Sousa, E., Vardasca, R., Teixeira, S., Seixas, A., Mendes, J., & Costa-Ferreira, A. (2017). A review on the application of medical infrared thermal imaging in hands. Infrared Physics & Technology, 85, 315-323. (not mine). This is a suggestion, but you can find several papers, and a review can be helpful. Please revise this part. Importantly, after this, you need to improve the hypotheses. Please, add more specific hypotheses.
Methods: The recruitment and inclusion/exclusion criteria were described in a good way.
Please, add more information about the therapy, the presence of atypical Parkinsonis, treatment with quetiapine (or similar, if present). I suppose that they were not under pharmacological wash-out.
2.2. Ten-Second Cold Water Stress Test (10sec-CWST): You must add more information about the infrared imaging data collection with the thermal camera. This is important for the replication of the study.
Similarly, more information needs to be added about the infrared data analysis, since it is not clear if you took into consideration specific regions of interest that you extracted with the software of the FLIR, or if you measured a single point. Moreover, E5 from FLIR is a quantitative camera, which measures the temperature in Celsius degrees, but you can also to calculate the Δ considering the first or best (based on a criterion) thermogram. This is important to study “the trend of cooling” in the two patient groups.
Results: You have also collected women, but it is not clear if you have checked for the Raynaud phenomenon, which is very common in women during fertile age and it is still less known. It should be interesting to see how many patients showed this when they were younger.
“ change in the clinical diagnosis from MSA to progressive supranuclear palsy within 5 years of disease onset.” This is interesting. I suppose that was also based on the “morning glory sign” in brain T1 MRI (?). It is only a clinical curiosity (sorry).
OSIT J and UPDRS need to be briefly described in the methods.
The AUC ROC results are interesting, and with more patients should be interesting to see if it can be >0.80.
Discussion: The noradrenergic explanation is interesting, but more evidences are needed. Please, add.
You need to add future directions to the section.
Author Response
Reply to Reviewer #1
We appreciate the useful comments and suggestions that have helped improve our manuscript. As indicated in these responses, we have considered all comments and suggestions while revising our manuscript.
Ten-Second Cold Water Stress Test Differentiates Parkinson’s Disease from Multiple System Atrophy: A Pilot Study
I have read the manuscript with interest, and you can find my appraisal, section by section, as follows:
As a general remark, I hope that my commentaries and appraisal can help improve the manuscript and give more insights.
Abstract: line 17: Please clarify the total sample size and the group size. This is not completely clear.
>Thank you for your comment. We have clarified the total sample size and group size on line 17-19.
Introduction: The section is well-written, but the symptoms of PD and MSA need to be described better at the start of the section.
>Thank you for your comment. We have modified the text to reflect this (line 37 and 42).
I also advise adding a brief description of the epidemiological data for PD and MSA, and adding related citations.
>Thank you for your comment. We have included this on line 35 and 40-41.
Moreover, I suggest adding the principal alterations that are possible to observe with MRI in MSA.
>Thank you for your comment. We have included this on line 47.
An important concern is about warm hands in MSA, since you have observed in your clinical practice, but you need to quantify this or report published data. Please, add.
>Thank you for your comment; however, we are sorry that we could not find any data to report.
Line 66: Since you study this phenomenon with infrared imaging, using a thermal camera, you need to improve the state of the art about previous studies that used this or a similar method/paradigm. You can find some information here: Sousa, E., Vardasca, R., Teixeira, S., Seixas, A., Mendes, J., & Costa-Ferreira, A. (2017). A review on the application of medical infrared thermal imaging in hands. Infrared Physics & Technology, 85, 315-323. (not mine). This is a suggestion, but you can find several papers, and a review can be helpful. Please revise this part.
>Thank you for providing this information. We have included this in the manuscript (line 71-73).
Importantly, after this, you need to improve the hypotheses. Please, add more specific hypotheses.
>Thank you for your comment. We have included this in the revised manuscript (line 73-75).
Methods: The recruitment and inclusion/exclusion criteria were described in a good way.
Please, add more information about the therapy, the presence of atypical Parkinsonis, treatment with quetiapine (or similar, if present). I suppose that they were not under pharmacological wash-out.
>Thank you for your comment. We have included further information (line 95-96).
2.2. Ten-Second Cold Water Stress Test (10sec-CWST): You must add more information about the infrared imaging data collection with the thermal camera. This is important for the replication of the study.
Similarly, more information needs to be added about the infrared data analysis, since it is not clear if you took into consideration specific regions of interest that you extracted with the software of the FLIR, or if you measured a single point.
>Thank you for your comment. We have included more information on this (line 108-109).
Moreover, E5 from FLIR is a quantitative camera, which measures the temperature in Celsius degrees, but you can also to calculate the Δ considering the first or best (based on a criterion) thermogram. This is important to study “the trend of cooling” in the two patient groups.
>Thank you for your comment. In this case, the measured temperature and delta were examined. Since delta showed a stronger difference, this was used. The measured temperature is shown in the supplementary figure.
Results: You have also collected women, but it is not clear if you have checked for the Raynaud phenomenon, which is very common in women during fertile age and it is still less known. It should be interesting to see how many patients showed this when they were younger.
>Thank you for your comment. I have included this on line 134-135.
“ change in the clinical diagnosis from MSA to progressive supranuclear palsy within 5 years of disease onset.” This is interesting. I suppose that was also based on the “morning glory sign” in brain T1 MRI (?). It is only a clinical curiosity (sorry).
>Thank you for your comment. The diagnosis was changed when progression revealed an increase in falls and frontal lobe signs. Furthermore, a head MRI was positive for the humming-bird sign and morning glory sign, in addition to midbrain atrophy.
OSIT J and UPDRS need to be briefly described in the methods.
>Thank you for your comment. This has been included on line 95 and 130.
The AUC ROC results are interesting, and with more patients should be interesting to see if it can be >0.80.
Discussion: The noradrenergic explanation is interesting, but more evidences are needed. Please, add.
>Thank you for your comment. No additional evidence was found that would be relevant to the current results. We have included a discussion on future studies and reports (line 225-227).
You need to add future directions to the section.
>Thank you for your comment. We have included this on line 259-265.

Reviewer 2 Report
Comments and Suggestions for Authors
Dear Authors,
Thank you for your engaging and informative manuscript. I appreciate the effort and clarity in your writing. However, I would like to raise a few important points for your consideration:
-
Sample Size Justification:
Please provide a clear explanation or calculation for the sample size. Justifying the sample size is essential to ensure the statistical power and reliability of your findings. -
Unequal Group Sizes:
It is unclear why the study groups have different numbers of participants. This discrepancy could introduce bias and affect the reliability of the MSA results. Please clarify the reason for this imbalance and how it was addressed analytically. -
CONSORT Guidelines:
Kindly include key elements from the CONSORT guidelines, such as details on randomization, blinding, and informed consent. These are critical to assess the internal validity and ethical standards of the study. -
Discussion of Future Applications:
It would strengthen the Discussion section to elaborate on the potential future applications of this method in clinical or research settings. This would add context to the significance of your work.
Thank you again for your valuable contribution.
Best regards,
Author Response
Reply to Reviewer #2
We appreciate the useful comments and suggestions that have helped improve our manuscript. As indicated in these responses, we have considered all comments and suggestions while revising our manuscript.
Dear Authors,
Thank you for your engaging and informative manuscript. I appreciate the effort and clarity in your writing. However, I would like to raise a few important points for your consideration:
- Sample Size Justification:
Please provide a clear explanation or calculation for the sample size. Justifying the sample size is essential to ensure the statistical power and reliability of your findings.
>Thank you for your comment. This study was a pilot study, and all patients who expressed their willingness to participate in the clinical study during the study period were tested. Based on the results, we hope to conduct this study in the future with a more defined sample size.
- Unequal Group Sizes:
It is unclear why the study groups have different numbers of participants. This discrepancy could introduce bias and affect the reliability of the MSA results. Please clarify the reason for this imbalance and how it was addressed analytically.
>Thank you for your comment. This study was a pilot study, and all patients who expressed their willingness to participate in the clinical study during the study period were tested. Based on the results, we hope to conduct this study in the future with a more defined sample size.
- CONSORT Guidelines:
Kindly include key elements from the CONSORT guidelines, such as details on randomization, blinding, and informed consent. These are critical to assess the internal validity and ethical standards of the study.
>Thank you for your comment. Since this research was a cross-sectional observational study, it was conducted in accordance with the STROBE guidelines. We have stated the study design in the title.
- Discussion of Future Applications:
It would strengthen the Discussion section to elaborate on the potential future applications of this method in clinical or research settings. This would add context to the significance of your work.
>Thank you for your comment. We have included this on line 259-265.

Round 2
Reviewer 1 Report
Comments and Suggestions for Authors
Ten-Second Cold Water Stress Test Differentiates Parkinson’s Disease from Multiple System Atrophy: A Pilot Study
The manuscript has been improved. However, as I requested in the previous report, you need to add specific information about the infrared protocol to allow for replicability. I perfectly know that there is no checklist for the studies that used infrared.
Here you can find some of the parameters that you need to add:
The distance between the hand and the camera in cm, the angle in degrees, and whether it is placed on a tripod.
Moreover, you need to add the sensor focal array size, NEDT (noise equivalent differential temperature), emissivity set at.. sampling rate (1 minute ), and the sensitivity of the camera.
Moreover, please add a figure and confirm if the ROI was circular (the diameter) and the ROI of reference. The method used to calculate the delta needs to be explained in a better way.
Please, add the information.
Author Response
Reply to Reviewer #1
We appreciate the useful comments and suggestions that have helped improve our manuscript. As indicated in these responses, we have considered all comments and suggestions while revising our manuscript.
The manuscript has been improved. However, as I requested in the previous report, you need to add specific information about the infrared protocol to allow for replicability. I perfectly know that there is no checklist for the studies that used infrared.
Here you can find some of the parameters that you need to add:
The distance between the hand and the camera in cm, the angle in degrees, and whether it is placed on a tripod.
>Thank you for your comment. We have included this on line 108-109.
Moreover, you need to add the sensor focal array size, NEDT (noise equivalent differential temperature), emissivity set at.. sampling rate (1 minute ), and the sensitivity of the camera.
>Thank you for your comment. We have included this on line 103-107.
Moreover, please add a figure and confirm if the ROI was circular (the diameter) and the ROI of reference.
>Thank you for your comment. We have included this on line 109-112 and supplementary figure 1.
The method used to calculate the delta needs to be explained in a better way.
>Thank you for your comment. We have included this on line 119-121.

Round 3
Reviewer 1 Report
Comments and Suggestions for Authors
The authors have addressed my last concern.